# MS785-MS27 Reactive Misfolded/Non-Native Zn-Deficient SOD1 Species Exhibit Cytotoxicity and Adopt Heterozygous Conformations in Motor Neurons

**DOI:** 10.3390/ijms25115603

**Published:** 2024-05-21

**Authors:** Eiichi Tokuda, Yume Sakashita, Naoya Tokoro, Ayano Date, Yasuhiro Kosuge, Tomohiro Miyasaka

**Affiliations:** 1Laboratory of Clinical Medicine, School of Pharmacy, Nihon University, 7-7-1 Narashinodai, Funabashi 274-8555, Chiba, Japan; 2Laboratory of Pharmacology, School of Pharmacy, Nihon University, 7-7-1 Narashinodai, Funabashi 274-8555, Chiba, Japan; kosuge.yasuhiro@nihon-u.ac.jp; 3Laboratory of Physiology and Anatomy, School of Pharmacy, Nihon University, 7-7-1 Narashinodai, Funabashi 274-8555, Chiba, Japan; miyasaka.tomohiro@nihon-u.ac.jp

**Keywords:** amyotrophic lateral sclerosis, MS785-MS27 antibody cocktail, Zn-deficient SOD1, recombinant proteins

## Abstract

Misfolding of superoxide dismutase-1 (SOD1) is a pathological hallmark of amyotrophic lateral sclerosis (ALS) with *SOD1* mutations. The development of antibodies specific for misfolded SOD1 deepens our understanding of how the protein participates in ALS pathogenesis. Since the term “misfolding” refers to various disordered conformers other than the natively folded one, which misfolded species are recognized by specific antibodies should be determined. Here, we molecularly characterized the recognition by MS785-MS27, an antibody cocktail experimentally confirmed to recognize over 100 ALS-linked SOD1 mutants. Indirect ELISA revealed that the antibody cocktail recognized Zn-deficient wild-type and mutated SOD1 species. It also recognized conformation-disordered wild-type and mutated SOD1 species, such as unfolded and oligomeric forms, but had less affinity for the aggregated form. Antibody-reactive SOD1 exhibited cytotoxicity to a motor neuron cell model, which was blocked by Zn treatment with Zn-deficient SOD1. Immunohistochemistry revealed antibody-reactive SOD1 mainly in spinal motor neurons of SOD1^G93A^ mice throughout the disease course, and the distribution after symptomatic stages differed from that of other misfolded SOD1 species. This suggests that misfolded/non-native SOD1 species exist as heterogeneous populations. In conclusion, MS785-MS27 recognizes various conformation-disordered SOD1 species lacking the Zn ion.

## 1. Introduction

Amyotrophic lateral sclerosis (ALS) is a neurodegenerative disease characterized by the loss of both upper and lower motor neurons in the central nervous system, resulting in muscle weakness, paralysis, and eventually death within 2 to 5 years of diagnosis [1]. While the majority of ALS cases (90% to 95%) are sporadic, the remaining cases are familial [1]. So far, over 30 causative genes have been identified in familial ALS [2], one of which is the gene encoding superoxide dismutase-1 (SOD1); mutations in this gene contribute to approximately 20% of familial cases [3] and more than 200 mutations have been found, which are collated in the ALSoD genetic database (http://alsod.iop.kcl.ac.uk/, accessed on 25 March 2024). The rates of disease progression in *SOD1*-linked familial ALS (*SOD1*-ALS) are divergent among the types of mutants. For example, A4V SOD1 causes a rapid disease progression with death after approximately 1 year of the disease onset [4], whereas D90A SOD1 leads to a slow disease progression with long survival times [5]. Wild-type SOD1 (WT SOD1) is an antioxidant that catalyzes the conversion of the superoxide anion to hydrogen peroxide and molecular oxygen [6]. However, the enzymatic activity of the ALS-linked SOD1 mutants varies depending on the variant, from 0% to 100% compared with that of WT SOD1 [7,8]. It is now accepted that the cytotoxicity of ALS-linked SOD1 mutants does not arise from a loss of their enzymatic activity [9]. A common property of ALS-linked *SOD1* mutations is that they compromise the stability of the protein’s quaternary structure, which promotes misfolding or disordered conformations [10,11]. Natively folded SOD1 is conformationally characterized by a homodimer containing one Cu and one Zn ion in each subunit, as well as by the formation of a disulfide bond between Cys57 and Cys146, which exhibits high thermostability (*T*_m_ > 90 °C) [12]. However, once SOD1 loses these metal ions, undergoes dimer dissociation, or undergoes cleavage of the disulfide bond, its stability is remarkably decreased. For instance, metal-free, disulfide bond-cleaved WT SOD1 exhibits lower thermostability (*T*_m_ = 42 °C) [12,13].

The concept that misfolded/non-native SOD1 participates in the pathogenesis of familial or even sporadic ALS has been reinforced by the development of antibodies specifically targeting these SOD1 species [14,15]. To our knowledge, more than 20 specific antibodies have been developed, a subset of which have the potential for use in therapeutic interventions [15,16,17,18]. Most specific antibodies of this kind are designed to bind to regions that are inaccessible in the natively folded form [19,20]. In other words, the specific antibodies recognize regions that are exposed at the surface only when SOD1 adopts a non-native conformation initiated by mutations or loss of cofactors involved in the conformational stability (e.g., Zn ions). Unlike for natively folded SOD1, it is difficult to define the inherent conformation of misfolded/non-native SOD1 due to the huge number of influential factors acting on this in combination (e.g., monomer/dimer, metal-free/bound, or disulfide bond cleavage/formation) [12,21]. Thus, misfolded/non-native SOD1 could consist of a heterogeneous population with distinct conformations. To better understand which misfolded/non-native SOD1 species participate in which disease process, it is critical to determine which misfolded/non-native SOD1 species are recognized by a specific antibody. However, the mechanisms behind the selectivity of most developed antibodies that specifically recognize misfolded/non-native SOD1 have not been well characterized at the molecular level. In most cases, they have just been determined by a comparison of specificity between natively folded and unfolded forms treated with chaotropic reagents (e.g., guanidinium ions) [22,23,24,25].

MS785 and MS27 are commercially available antibodies specifically targeting misfolded SOD1 [26,27]. These antibodies are designed on the basis of structural information about a region where all ALS-linked SOD1 mutants bind to Derlin-1, a component of the endoplasmic reticulum-associated degradation machinery [26,27]. Although each of these antibodies fails to recognize ALS-linked SOD1 mutants with a point mutation in their epitope region (amino acids 8–14 for MS785; amino acids 30–40 for MS27), the use of an MS785 and MS27 cocktail compensates for each of their shortcomings, with this mixture recognizing over 100 ALS-linked SOD1 mutants, as determined by immunoprecipitation [27]. Thus, the MS785-MS27 cocktail is the only antibody in which the affinity for a variety of ALS-linked SOD1 mutants has been experimentally confirmed. According to previous cell-based studies [27,28], this antibody cocktail recognizes SOD1 species that show an increase in level upon the treatment of cells with a Zn-chelating reagent, implying that the antibody cocktail recognizes “Zn-deficient SOD1 species”. However, despite substantial efforts, it is unclear which misfolded/non-native SOD1 species can be recognized by this antibody cocktail at the molecular level.

Against the above background, the aim of the present study is to molecularly characterize the recognition by the MS785-MS27 antibody cocktail of various SOD1 species with altered conformational integrity.

## 2. Results

### 2.1. MS785-MS27 Antibody Cocktail Recognizes WT SOD1 Species Lacking the Incorporated Zn Ion

To promote rapid analyses of the SOD1 species recognized by the MS785-MS27 antibody cocktail, we developed an indirect enzyme-linked immunosorbent assay (ELISA) coupled with this cocktail. In our ELISA, we used metal-free, disulfide bond-cleaved WT SOD1 (apo-WT SOD1^SH^) as a representative species of Zn-deficient SOD1, as well as an internal standard for comparisons among different 96-well plates. We first assessed the consistency and reproducibility of our ELISA. Apo-WT SOD1^SH^ with high purity, as confirmed by sodium dodecyl sulfate-polyacrylamide gel electrophoresis (SDS-PAGE; Figure 1A) or vehicle buffer, was added to 48 wells in the plate and detected by the incubation of the MS785-MS27 antibody cocktail followed by the use of a horseradish peroxidase (HRP) colorimetric system. The ELISA signal intensity of apo-WT SOD1^SH^ was 1.08 ± 0.05 (mean ± SD), whereas that of the vehicle buffer was 0.06 ± 0.01. Thus, the coefficient of variation (CV) value in our ELISA was 4.5%, which was determined by dividing the SD value by the mean value in apo-WT SOD1^SH^ (Figure 1B). Furthermore, the ratio of the signal intensity to the background (S/B) in the ELISA was 19 (Figure 1B), which was calculated by dividing the mean value of apo-WT SOD1^SH^ by that of the vehicle buffer. These results indicate that our ELISA achieved not only small variation among samples (CV < 10%), but also high affinity for apo-WT SOD1^SH^ accompanied by a low background (S/B > 3.0). Moreover, the *Z*′-factor, an indicator of assay quality, was 0.84 (Figure 1B), implying that our ELISA is suitable for high-throughput analyses because the *Z*′-factor exceeds the established threshold of 0.50 [29].

We confirmed the antibody cocktail recognition for apo-WT SOD1^SH^ by competitive ELISA. The principle of which involves a competitive binding process between a sample antigen and a reference antigen. The primary antibody is incubated with the sample antigen, and then, the antibody–antigen complexes are added to 96-well plates that have been precoated with the reference antigen. When the sample antigen is abundantly bound to the antibody, little free antibody exists and there is less binding to the reference antigen coating the plates. In the present study, the antibody cocktail at 1 μg/mL was mixed with apo-WT SOD1^SH^ at different concentrations. The complex of the antibody cocktail and apo-WT SOD1^SH^ was used as a primary antibody in the indirect ELISA. As expected, preincubation of the antibody cocktail with apo-WT SOD1^SH^ led to a decrease in the ELISA signals in a concentration-dependent manner (Figure 1C). Taking the findings together, we succeeded in developing an indirect ELISA coupled with the MS785-MS27 antibody cocktail exhibiting high detection sensitivity as well as a low background.

Wild-type SOD1 contains one Cu ion and one Zn ion in its monomeric subunit [12]. Thus, we next examined the antibody cocktail’s recognition of WT SOD1 into which these metals were differently incorporated. We treated apo-WT SOD1^SH^ with 1 molar equivalent of either Cu or Zn ions, and investigated the antibody cocktail’s recognition of the metal-bound forms using indirect ELISA. We found that the antibody cocktail recognized Cu-WT SOD1^SH^ (Zn-deficient form), whereas it did not recognize Zn-WT SOD1^SH^ (Cu-deficient form) (Figure 1D). Moreover, the antibody cocktail did not react with Cu/Zn-WT SOD1, the natively folded and enzymatically active form (Figure 1D), indicating that the antibody cocktail recognizes Zn-deficient WT SOD1^SH^ (metal-free or Cu-binding form).

To determine the role of the Zn coordination to SOD1 in the antibody cocktail’s recognition, we treated apo-WT SOD1^SH^ with 1 to 4 molar equivalents of Zn ions. As shown in Figure 1E, the antibody cocktail reactivity was significantly decreased in a dose-dependent manner upon the Zn coordination to SOD1. This decrease was not observed when the Zn binding site of SOD1, namely, residues His63, His71, His80, and Asp83, was lost by amino acid substitutions (Figure 1E, red). We also performed similar experiments using disulfide bond-forming WT SOD1 (SOD1^S-S^) with different incorporated metals. The antibody cocktail retained reactivity to Zn-deficient SOD1^S-S^ (Figure 1F). Taking these findings together, the MS785-MS27 antibody cocktail discriminates conformational changes related to the lack of Zn ion incorporation in SOD1 regardless of whether the protein forms a disulfide bond.

### 2.2. MS785-MS27 Antibody Cocktail Reacts with Various Conformation-Disordered WT SOD1 Species

Conformation-disordered SOD1 species such as unfolded, oligomeric, and aggregated forms are pathological hallmarks of *SOD1*-ALS or even of a subset of sporadic ALS [9,23,30,31,32]. We determined whether the MS785-MS27 antibody cocktail could recognize conformation-disordered WT SOD1 species. We prepared unfolded, oligomeric, and aggregated forms of SOD1 in a test tube as sources of apo-WT SOD1^S-S^, as described previously [31,33], and analyzed the antibody cocktail recognition of these disordered species using indirect ELISA. The antibody cocktail strongly reacted with both unfolded and oligomeric SOD1, and the reactivities were of the same intensity as those for apo-WT SOD1^SH^ (Figure 2A). Meanwhile, the antibody cocktail showed less affinity for SOD1 aggregates (Figure 2A, *p* < 0.01 vs. apo-WT SOD1^SH^). This was unsurprising because the epitopes of MS785 and MS27 are amino acids corresponding to residues 8–14 and 30–40 in human SOD1, respectively, which are regions buried within aggregates [34,35]. Such lower reactivity toward SOD1 aggregates is unrelated to them being difficult to attach to or easily detached from the 96-well plate (Figure 2B, *p* = 0.54) because they were detected at the same intensity as other SOD1 species by an anti-SOD1 antibody raised against full-length human SOD1 and reacted with all SOD1 conformers [31].

We also characterized the recognition by MS785 or MS27 alone of various WT SOD1 species including metal binding forms, as well as conformation-disordered forms. Indirect ELISA revealed that MS785 or MS27 alone had recognition patterns and affinities for various WT SOD1 species similar to those of the MS785-MS27 antibody cocktail, while the MS27 alone strongly responded to unfolded WT SOD1^SH^ (Appendix A). Overall, these findings indicate that the antibody cocktail has a broad response to conformation-disordered SOD1 species lacking the incorporated Zn ion.

### 2.3. MS785-MS27 Recognizes Zn-Deficient SOD1 with ALS-Linked Mutations

Based on previous cell-based studies [27,28], the MS785-MS27 antibody cocktail recognizes over 100 ALS-linked SOD1 mutants. To further molecularly characterize whether the antibody cocktail can recognize ALS-linked SOD1 mutants with distinct biopsychical properties and conformations, we used seven different types of the mutants classified into three groups: (i) WT-like mutants, G37R [7], D90A [36], and G93A [37]; (ii) dimer interface mutants, A4V [38] and I113T [39]; and (iii) metal binding region mutants, H46R and G85R [7,40] (Figure 3A). We first investigated the antibody cocktail recognition of apo-SOD1^SH^ with these mutations, the purity of which was confirmed by SDS-PAGE (Figure 3B). All tested apo-SOD1^SH^ with ALS-linked mutations, except A4V and G37R, exhibited the same signal intensity as that of WT SOD1 (Figure 3C). The antibody cocktail had 30% less affinity for A4V SOD1 than for WT SOD1 (Figure 3C), which was consistent with the findings of a previous cell-based study [27]. Apo-G37R SOD1^SH^ had as much as half of the signal intensity compared with the other mutants and WT SOD1 (Figure 3C). This was reasonable because G37R SOD1 has a point mutation in the epitope region of MS27 (^30^KVWGSIK**G**LTE^40^; the amino acid substitution G37R of SOD1 is shown in bold) [27]. Thus, the reactivity of the antibody cocktail to apo-G37R SOD1^SH^ stems from MS785, but not MS27. Indeed, MS785 alone recognized apo-G37R SOD1^SH^ at a similar intensity to the apo-WT SOD1^SH^, whereas MS27 alone did not (Appendix A). The antibody cocktail exhibited similar affinity patterns to apo-SOD1^S-S^ with the ALS-linked mutations (Figure 3D).

We also determined whether the antibody cocktail could recognize conformation-disordered SOD1 with ALS-linked mutations, including unfolded, oligomeric, and aggregated forms. The antibody cocktail had similar affinity to the unfolded SOD1 mutants (Figure 3E), except A4V and G37R, as to unfolded WT SOD1 (Figure 2A). The antibody cocktail also showed similar affinity and intensity to oligomeric SOD1 mutants (Figure 3F), except A4V and G37R, as to oligomeric WT SOD1 (Figure 2A). By contrast, the antibody cocktail had lower affinity to mutant SOD1 aggregates (Figure 3G), which was consistent with the findings for WT SOD1 aggregates (Figure 2A).

To address whether the Zn coordination to the ALS-linked SOD1 mutants plays a major role in the antibody cocktail recognition, we treated the seven different types of SOD1 mutants with 1 to 4 molar equivalents of Zn ions, and analyzed the antibody cocktail recognition of these mutants with distinct incorporation of Zn ions using indirect ELISA. Regarding the WT-like mutants (G37R, D90A, and G93A), as well as dimer interface mutants (A4V and I113T), the antibody cocktail reactivity significantly decreased in a concentration-dependent manner upon the Zn treatment (Table 1), suggesting that the Zn coordination to these mutants is associated with conformational changes, possibly incomplete folding, which inhibits recognition by the antibody cocktail. However, focusing on these Zn-treated mutants, the antibody cocktail still had higher affinity for these mutants than for WT SOD1 (Table 1), implying that these mutants decrease the capacity to incorporate a Zn ion. With respect to the metal binding region mutants (H46R and G85R), the antibody cocktail still recognized the mutants even upon Zn treatment (Table 1). Taken together, these results indicate that the antibody cocktail recognizes various conformation-disordered SOD1 species lacking the incorporated Zn ion with ALS-linked mutations, and that the ALS-linked SOD1 mutants exhibit significantly decreased affinity for Zn ions.

### 2.4. SOD1 Species Lacking the Incorporated Zn Ions Have Cytotoxic Effects on NSC-34 Cells

To determine whether the MS785-MS27-reactive SOD1 species could exhibit toxic effects on cultured cells, we first investigated the proliferation of NSC-34 cells, a hybridoma with mouse spinal motor neurons and neuroblastoma cells, by measuring the metabolic activity using Cell Counting Kit-8. The cells were treated with apo-WT SOD1^SH^ or apo-SOD1^SH^ that lacks the Zn binding site, as representatives of MS785-MS27-reactive SOD1 species (Figure 1E), at different concentrations for 48 h. We found that the cell proliferation was significantly decreased in NSC-34 cells treated with apo-WT SOD1^SH^ at the concentration range of 0.25 to 10 μM, compared with that of vehicle controls (Figure 4A, black). Notably, the proliferation of NSC-34 treated with apo-SOD1^SH^ that lacks the Zn binding site decreased markedly compared with that of apo-WT SOD1^SH^ (Figure 4A, red). For example, apo-WT SOD1^SH^ exhibited an IC_50_ value of 10 μM for cell proliferation, whereas apo-SOD1^SH^ that lacks the Zn binding site exhibited one of 1 μM (Figure 4). We confirmed these results via a lactate dehydrogenase (LDH) cytotoxicity assay, measuring the activity of LDH released from damaged cells. Both SOD1 species still exhibited cytotoxicity toward the NSC-34 cells, and apo-SOD1^SH^ that lacks the Zn binding site had 10-fold greater cytotoxicity than apo-WT SOD1^SH^ (EC_50_ for apo-WT SOD1^SH^: 10 μM, EC_50_ for apo-SOD1 that lacks the Zn binding site: 1 μM, Figure 4B). These cytotoxic effects were blocked by the pretreatment of apo-WT SOD1^SH^ with Zn ions (Figure 4C,D, black). However, this protection was not observed in SOD1^SH^ that lacks the Zn binding site (Figure 4C,D, red), indicating that the cytotoxic effects of apo-SOD1^SH^ are related to lack of Zn ion incorporation in the protein. In summary, the MS785-MS27-reactive SOD1 species exhibit cytotoxicity toward a cellular model of motor neurons, and their cytotoxic effects could be rescued by additional Zn treatment for WT SOD1 species.

### 2.5. Distribution of MS785-MS27-Reactive SOD1 Species in the Spinal Cord of G93A SOD1 Mice

To clarify whether the MS785-MS27-reactive SOD1 species could localize in motor neurons and how their distribution changes during the disease course of a mouse model of *SOD1*-ALS, we performed immunofluorescence with the antibody cocktail on the lumbar spinal cord from high-copy G93A SOD1 mice at different disease stages. At the presymptomatic stage (60 days of age), the MS785-MS27-reactive SOD1 species were present as small granules in motor neurons (Figure 5A, inset), and appeared as diffuse staining in axons (Figure 5A, arrows). After the onset of the symptoms (90 days of age), the SOD1-positive granules observed in motor neurons became larger (Figure 5B, inset), and were located around vacuoles in the ventral horn (Figure 5B, arrows). At the terminal stage (120 days of age), the SOD1 species were still detectable in motor neurons (Figure 5C, inset) and exhibited fibril-like structures outside the motor neurons (Figure 5C, arrows).

Notably, in mice overexpressing human WT SOD1 at 120 days of age, the SOD1 species were also observed in motor neurons (Figure 5D, inset) and vacuoles (Figure 5D, arrows), suggesting that WT SOD1 could lack Zn ion incorporation in the protein and undergo a conformational shift to a non-native state in vivo. By contrast, no fluorescence signals were detected in the ventral horn of non-transgenic mice (Figure 5E). We sought to confirm the recognition of murine SOD1 by the MS785-MS27 antibody cocktail by indirect ELISA with recombinant murine SOD1 protein, the high purity of which was confirmed by SDS-PAGE (Appendix A). Indirect ELISA demonstrated that the antibody cocktail reacted with the metal-free form as well as the unfolded form of murine SOD1, but not the natively folded form (Appendix A). MS27 alone did not recognize murine SOD1 species with which the antibody cocktail reacted (Appendix A). Thus, the reactivity of the MS785-MS27 antibody cocktail to metal-free or unfolded murine SOD1 appears to come from MS785. This is reasonable because the sequence of murine SOD is 100% identical to that of human SOD1 in the epitope region of MS785, whereas there is 54.5% homology between the murine and human SOD1 sequences in the epitope region of MS27 (Appendix A). Taking these findings together, the endogenous murine SOD1 in non-transgenic mice exists in a conformation that the MS785-MS27 antibody cocktail does not recognize, presumably the natively folded form [41,42].

Taking into account the presence of the MS785-MS27-reactive SOD1 species in cell types other than motor neurons throughout the disease course of G93A SOD1 mice (Figure 5A–C), we investigated whether the SOD1 species could localize in astrocytes, microglia, or both. Immunofluorescence studies demonstrated that the SOD1 species localized in a subset of microglia at all stages of the disease (Figure 5I–K), whereas the SOD1 species were not found in astrocytes (Figure 5F–H). Taking these findings together, the major site of the MS785-MS27-reactive SOD1 species in the ventral horn is the motor neurons over the disease course of G93A SOD1 mice.

### 2.6. MS785-MS27-Reactive SOD1 Species Are Differentially Distributed from the Known Misfolded/Non-Native SOD1 Species in G93A SOD1 Mice

Previous immunohistochemical studies using a panel of specific antibodies for misfolded/non-native SOD1 demonstrated that misfolded/non-native SOD1 species likely exist as a heterogeneous population in vivo [25]. In other words, misfolded/non-native SOD1 could exhibit divergent conformations in the tissues affected by disease, and they could have different implications for the disease pathogenesis. To address the implications of the MS785-MS27-reactive SOD1 species among various misfolded/non-native conformers, we performed immunofluorescence on the lumbar spinal cord using the MS785-MS27 antibody cocktail and EDI, a commercially available antibody that specifically recognizes misfolded/non-native SOD1 for which the epitope is exposed at the dimer interface [14]. At the presymptomatic stage, the distribution of the MS785-MS27-reactive SOD1 species was identical to that of EDI-reactive misfolded/non-native SOD1 (Figure 6A), showing that both species were mainly observed in motor neurons. After the onset of symptoms, EDI-reactive misfolded/non-native SOD1 disappeared from the motor neurons, whereas the MS785-MS27-reactive SOD1 species were still located in them (Figure 6B). In contrast to the motor neurons, both species were colocalized in the vacuoles (Figure 6B, arrows) as well as in ventral root axons (Figure 6B). These manifestations continued at the terminal stage of the disease, demonstrating that both species were present as fibril-like structures (Figure 6C, arrows). Taking these findings together, there are differences in spatial distribution between MS785-MS27-reactive SOD1 and other species during the disease course of G93A SOD1 mice. Thus, the population of the misfolded/non-native SOD1 species could be homozygous at the presymptomatic stage, whereas after the disease onset, the misfolded/non-native SOD1 could adopt divergent conformations.

## 3. Discussion

### 3.1. Common Conformational Features of the MS785-MS27-Reactive Misfolded/Non-Native SOD1 Species

The present study sheds light on the conformational features of the MS785-MS27-reactive SOD1 species, both WT and the ALS-linked SOD1 mutants. Here, we can speculate about the possible conformation that is shared among the MS785-MS27-reactive SOD1 species with regard to the epitope region of each antibody. The epitope region of MS785 is the peptide corresponding to amino acids 8 to 14 in human SOD1 [26], which is located in the small loop between the β_1_ and β_2_ strands. Meanwhile, the epitope region of MS27 is the peptide corresponding to amino acids 30 to 40 in human SOD1 [27], which covers the β_3_ strand, followed by a small loop. It is possible that the apo-form, unfolded form, and oligomeric form share a common disordered conformation around the β_1_ to β_3_ strands. In support of this idea, the X-ray crystal structure of apo-SOD1 revealed that the regions of the β_1_ to β_3_ strands are exposed to the solvent, in contrast to the case for the natively folded SOD1 (PDB: 3ECU), which provokes the misfolding of SOD1 [43]. Notably, almost all conformation-disordered WT SOD1 species, with the exception of unfolded SOD1^SH^, are equally recognized by MS785 or MS27 alone (Appendix A), which is in agreement with the recognition patterns by the MS785-MS27 antibody cocktail (Figure 2 and Figure 3). Thus, the exposed regions of β_1_ to β_3_ in SOD1 are likely to be targets for pharmacotherapy to correct disordered conformations of misfolded/non-native SOD1 species.

### 3.2. Implications of MS785-MS27-Reactive SOD1 Species for Cellular Events Related to ALS Pathogenesis

Although MS785- or MS27-reactive SOD1 species are reported to exist in the lumbar spinal cords of terminally ill G93A SOD1 mice [27], the cell types in the spinal cords of these species and how the distribution of SOD1 species changes during the disease course are unknown. In this study, we provide evidence that the MS785-MS27-reactive SOD1 species were mainly localized in the motor neurons throughout the disease course (Figure 5A–C), which suggests that the SOD1 species existing in motor neurons are deficient in Zn ions. These findings are consistent with a recent study [44] showing that 42% of human ALS cases exhibit Zn deficiency in WT or mutated SOD1 localized in the ventral horn. How do the MS785-MS27-reactive SOD1 species existing in the motor neurons contribute to the pathogenesis of ALS? The antibody cocktail-reactive apo-WT SOD1^SH^ and apo-SOD1^SH^ that lacks the Zn binding site exhibited toxic effects on the NSC-34 cells (Figure 4A,B). Our results are consistent with previous cell-based studies, demonstrating that apo-SOD1^S-S^ decreases the proliferation in several cell lines as neuronal models and that its toxic effect is rescued by treatment with Zn ions for the SOD1 species [45,46]. In terms of the molecular mechanisms underlying the toxic effects of apo-SOD1 species, it has been reported that the species provoke endoplasmic reticulum stress and activate its downstream signaling for apoptosis [47]. Interestingly, the endoplasmic reticulum stress-related cell death is induced by the dysregulation of intracellular Zn homeostasis [28]. Thus, apo-SOD1 species, positive for the MS785-MS27 antibody cocktail, appear to induce cell death via the endoplasmic reticulum in a manner related to Zn dysregulation.

As mentioned above, it would be beneficial to decrease the amounts of Zn-deficient SOD1 species in order to protect motor neurons. One approach to achieve this is to perform Zn coordination to SOD1 species. An earlier in vivo study demonstrated that treatment with Zn^II^ (atsm), a small compound that coordinates Zn ions, improved motor functions of different mouse lines of *SOD1*-ALS [48]. We have also shown that metallothionein-I, a major Zn binding protein in the cells, suppresses the loss of motor neurons in G93A SOD1 mice [49,50]. Thus, restoration of the Zn bioavailability in SOD1 itself or in the microenvironment surrounding SOD1 would be important for the correction of SOD1 misfolding and suppression of motor neuron death. We propose that the reactivity of the MS785-MS27 antibody cocktail could be used as an indicator of Zn bioavailability in SOD1, since its recognition negatively correlates with the incorporation of Zn ions in SOD1 (Figure 1E and Table 1).

### 3.3. Application of the MS785-MS27 Antibody Cocktail for Immunotherapies Targeting Misfolded/Non-Native SOD1

By using the co-immunostaining with the MS785-MS27 antibody cocktail and EDI, we provide evidence for there being divergent populations of misfolded/non-native SOD1 species that dynamically change over the disease course of G93A SOD1 mice. While the conformation of misfolded/non-native SOD1 at the early stage is likely to be homogeneous (Figure 6A), SOD1 species appear to adopt various conformations as the disease progresses (Figure 6B,C). Our histological data are supported by a previous study reporting that there were distinct distribution patterns of misfolded/non-native SOD1 species when these species were immunolabeled with a panel of specific antibodies [25]. The fact that the conformations of misfolded/non-native SOD1 species become diverse during the disease course should be kept in mind when passive immunization targeting misfolded/non-native SOD1 species is designed. In this respect, SOD1 aggregates are a good example of the difficulty in designing passive immunization. Immunohistochemical analyses with several lines of *SOD1*-ALS mice have indicated that the temporal pattern of initiation of SOD1 aggregation differs in each mutant [41,51,52]. While G85R SOD1 aggregates start to accumulate in the spinal cord of G85R SOD1 mice after onset of the disease [52], passive immunization with a specific antibody for SOD1 aggregates does not rescue the disease phenotype of G85R SOD1 mice, even though the immunization begins before the start of SOD1 aggregation [17]. By contrast, the aggregate-specific antibody attenuates the disease phenotype of G85R SOD1 mice when the mice are inoculated with seeded G85R SOD1 aggregates beforehand [17]. Thus, the conformations of SOD1 aggregates appear to be distinct between spontaneously accumulated aggregates and cell-to-cell transmitted ones, and whether outcomes of passive immunization could be improved depends on the conformations of SOD1 species, even using the same antibody. Based on our findings that the MS785-MS27 antibody cocktail mainly recognizes soluble misfolded/non-native SOD1 species (Figure 2A and Figure 3), which are observed in G93A SOD1 mice at the presymptomatic stage (Figure 5), we propose that the antibody cocktail might delay the disease onset in the mice.

### 3.4. Applications of the MS785-MS27 Antibody Cocktail for Diagnosing ALS

One of the major issues in clinical research on ALS is that its diagnosis is often delayed due to the need to rule out other neurological diseases with similar symptoms [53]. Thus, the establishment of a simple test that can definitively diagnose ALS is eagerly anticipated. In particular, biomarkers to predict disease development or progression would be required for presymptomatic sporadic ALS cases with no genetic predisposition [53]. We recently reported that cerebrospinal fluid from sporadic ALS cases contains various conformation-disordered WT SOD1 species [54], which opens up new avenues for the use of conformation-disordered WT SOD1 species as biomarkers for the diagnosis of sporadic ALS. With respect to the potential of MS785-MS27 for the diagnosis of ALS, a previous study demonstrated that a sandwich ELISA with MS785 alone discriminates familial ALS cases carrying *SOD1* mutations from non-neurological controls using B-lymphocytes [27]. Considering that the MS785-MS27 antibody cocktail recognizes various conformation-disordered WT SOD1 species lacking the incorporation of Zn ions for both in vitro (Figure 2) and in vivo samples (Figure 5D), the antibody cocktail could be used for the detection of misfolded/non-native WT SOD1 species in biomaterials from sporadic ALS cases. Future studies should examine whether the MS785-MS27-reactive SOD1 species could exist in cerebrospinal fluid from sporadic ALS cases and, if so, how the amounts of SOD1 species could change over the disease course.

## 4. Materials and Methods

### 4.1. Expression, Purification, and Demetallation of Recombinant SOD1 Proteins

A bacterial expression plasmid, pET28a(+), encoding human *SOD1* with I113T that was N-terminally tagged with hexahistidine, was purchased from Addgene (#117703; Watertown, MA, USA). Plasmids encoding WT, A4V, G37R, H46R, G85R, D90A, or G93A human *SOD1* gene were prepared using site-directed mutagenesis, and correct sequences of each construct were confirmed by Azenta (Tokyo, Japan). Human *SOD1* with mutations in the Zn binding sites was designed from the WT sequence by replacement of residues H63A, H71A, H80G, and D83A, and subcloned into the pET28a(+) vector. A bacterial expression plasmid, pET28a(+), encoding mouse WT *Sod1* that was N-terminally tagged with hexahistidine, was also obtained from Addgene (#117701). Each expression vector was transformed into SHuffle *E. coli* (New England Biolabs, Ipswich, MA, USA). Colonies isolated from the LB-agar plate were inoculated onto LB medium with 100 μg/mL kanamycin sulfate at 200 rpm and 30 °C for 14 h. Bacterial overexpression was initiated by adding the preculture medium to 500 mL of the LB medium. When the turbidity (OD_600_) of the *E. coli* reached between 0.60 and 0.80, the cells were cultured with 0.5 mM isopropyl-β-*D*-thiogalactopyranoside (Bio Medical Science, Tokyo, Japan) at 200 rpm and 20 °C for 20 h. The *E. coli* was lysed by sonication (Sonifier SFX250; Branson Emerson, Brookfield, CT, USA) in a buffer (pH 7.0) containing 2% (*v*/*v*) Triton X-100, 50 mM Tris, 500 mM NaCl, EDTA-free Complete Protease Inhibitor Cocktail (Merck, Darmstadt, Germany), 1 U DNase I (Fujifilm Wako Pure Chemical Corporation, Osaka, Japan), and 5 mM MgSO_4_. The cell lysates were centrifuged at 20,000× *g* and 4 °C for 30 min, and the supernatant was filtered with a 0.22 μm syringe-driven filter unit (Millipore, Burlington, MA, USA). The SOD1 with a hexahistidine tag was isolated using a Ni^2+^ affinity chromatograph (Complete His-Tag Purification Column; GE Healthcare, Chicago, IL, USA) by an ÄKTA Start (Cytiva, Tokyo, Japan). Pre-equilibration of the column was carried out using a buffer (pH 7.0) containing 50 mM Tris, 1 M NaCl, and 50 mM imidazole. Elution of the fusion protein was performed with a buffer (pH 7.0) containing 50 mM Tris, 100 mM NaCl, and 250 mM imidazole.

The SOD1 with a hexahistidine tag was demetallated by dialysis in a Spectra/Por^®^ (molecular mass cut-off: 6–8 kDa; Repligen, Waltham, MA, USA) against a buffer (pH 4.0) containing 50 mM sodium acetate, 100 mM NaCl, and 5 mM EDTA at 4 °C for 20 h. The apo-SOD1 with a hexahistidine tag was neutralized by dialysis with a buffer (pH 7.0) containing 50 mM Tris, 100 mM NaCl, and 5 mM EDTA for 4 h at 4 °C. The hexahistidine tag was cleaved by treatment with 4 U thrombin, a serine protease (GE Healthcare), at 20 °C for 20 h. The non-cleaved fusion protein and thrombin were removed using a Complete His-Tag Purification Column and Benzamidine Column (Cytiva), respectively. The SOD1 was further purified by size exclusion chromatography (Superdex™ 200 Increase Column 10/300 GL; Cytiva) with an ÄKTA Go (Cytiva). The concentration of SOD1 protein was determined by an Eppendorf BioSpectrometer (Eppendorf, Hamburg, Germany) using a monomeric molar extinction coefficient at 280 nm of 5500 M^−1^ cm^−1^. The purity of SOD1 was confirmed by SDS-PAGE followed by InstantBlue^®^ Coomassie Protein Stain (Abcam, Cambridge, UK).

### 4.2. Preparation of Metal Binding Forms and Conformation-Disordered SOD1

Metal binding forms of WT SOD1 were prepared as described elsewhere with slight modification [31,33]. A buffer (pH 7.0) containing 50 mM Tris and 100 mM NaCl was treated with a Chelex 100 Resin (100–200 mesh; Bio-Rad, Hercules, CA, USA) to remove divalent ions. Apo-WT SOD1^S-S^ was treated with 10 mM (±) dithiothreitol (DTT) at 37 °C for 1 h to obtain the disulfide bond-cleaved form. DTT was removed using a Zeta Spin Desalting Column (molecular mass cut-off: 7 kDa; Thermo Fisher Scientific, Waltham, MA, USA), in accordance with the manufacturer’s protocol. Apo-WT SOD1^SH^ was treated with 1 to 4 molar equivalents of Zn or Cu ions at 4 °C for 3 h. The Cu- and Zn-bound form of WT SOD1 was purchased from Sigma-Aldrich (S9636; St. Louis, MO, USA), which was isolated from human erythrocytes and used as a natively folded form (also known as an enzymatically active form).

Conformation-disordered SOD1 species, including unfolded, oligomeric, and aggregated forms, were prepared as described previously with slight modification [31,33]. For preparation of the unfolded form, Apo-SOD1^S-S^ at 100 μM was treated with 10 mM DTT at 37 °C for 1 h. After removing the DTT using a Zeta Spin Desalting Column (molecular mass cut-off: 7 kDa; Thermo Fisher Scientific), apo-SOD1^SH^ was treated with a buffer (pH 7.0) containing 50 mM Tris, 100 mM NaCl, 5 mM EDTA, and 6 M guanidine hydrochloride at 37 °C for 2 h. For preparation of the oligomeric form, apo-SOD1^S-S^ at 100 μM was incubated at 37 °C for 5 days to induce non-native cross-linking between Cys residues in SOD1 [55]. For preparation of the aggregated form, apo-SOD1^SH^ at 100 μM was added into a ProteoSave 96-well plate (Sumitomo Bakelite Co., Ltd., Tokyo, Japan) and agitated in the presence of a plastic POM ball (*ϕ*2.4 mm; Sanplatec Co., Ltd., Tokyo, Japan) at 1200 rpm and 37 °C for 5 days. The suspension of the SOD1 aggregates was collected and subjected to centrifugation at 20,000× *g* for 30 min at 4 °C. The supernatant was discarded and the pellet was resuspended in a buffer (pH 7.0) containing 50 mM Tris, 100 mM NaCl, and 5 mM EDTA. The SOD1 aggregates were sonicated using a Handy Sonic (UR-21P; Tony Seiko Co., Ltd., Tokyo, Japan). The concentration of SOD1 aggregates was determined by an Eppendorf BioSpectrometer, using a monomeric molar extinction coefficient at 280 nm of 5500 M^−1^ cm^−1^.

### 4.3. ELISA

To analyze the conformation-disordered SOD1 species, EDTA at a final concentration of 5 mM was added to Tris-buffered saline (TBS, pH 7.4) or TBS with 0.05% (*v*/*v*) Tween 20 (TBST, pH 7.4) to avoid the metal re-coordination to SOD1. For experiments on the metal binding forms of SOD1, TBS or TBST was treated with a Chelex 100 Resin (100–200 mesh; Bio-Rad) to remove divalent metal ions. SOD1 species at 5 μM were coated onto a Nunc Immuno MaxiSorp 96-well plate (Thermo Fisher Scientific) at 4 °C for 18 h. The plates were blocked with 3% (*w*/*v*) bovine serum albumin in TBS (pH 7.4) at room temperature for 1 h. After three washes with TBST, primary antibodies, MS785-MS27 cocktail (0.025 μg/mL, FDV-0021A; Funakoshi, Tokyo, Japan), MS785 (0.025 μg/mL, FDV-0021B; Funakoshi), MS27 (0.025 μg/mL, FDV-0021C; Funakoshi), or pan-SOD1 (0.5 μg/mL, sc-11407; Santa Cruz Biotechnologies, Dallas, TX, USA), were added, and the plates were incubated at 4 °C for 18 h. After three washes with TBST, HRP-conjugated secondary antibody against rat IgG (1:1000, AS028; AB Clonal, Woburn, MA, USA) or rabbit IgG (1:1000, A9169; Sigma-Aldrich) was added, and the plates were incubated at room temperature for 1 h. As a chromogenic substrate, 100 mM citrate buffer (pH 5.0) containing 0.03% (*v*/*v*) hydrogen peroxide and 1 mg/mL *O*-phenylenediamine dihydrochloride (Fujifilm Wako Pure Chemical Corporation) was added, and the color reaction was terminated by treatment with 1 M hydrochloric acid. The absorbance at 490 nm was measured on a microplate reader (FLUOstar Omega; BMG LABTECH, Ortenberg, Germany).

For competitive inhibition of ELISA, the MS785-MS27 antibody cocktail at 1 μg/mL was mixed with apo-WT SOD1^SH^ over a concentration range of 10^−4^ to 10^1^ μM at 4 °C for 18 h. The complex of MS785-MS27 and apo-WT SOD1^SH^ was used as a primary antibody for indirect ELISA, where apo-WT SOD1^SH^ at 5 μM was coated onto the Nunc Immuno 96-well plate. The subsequent procedures of competitive ELISA were the same as mentioned above for the indirect ELISA.

### 4.4. Assessment of Cell Proliferation and Cytotoxicity

The culture and maintenance of NSC-34 cells were performed as described previously [54,56]. In brief, NSC-34 cells were cultured in Dulbecco’s modified Eagle’s medium (DMEM) containing 4.5 g/L *D*-glucose (Nacalai Tesque, Kyoto, Japan) supplemented with 10% (*v*/*v*) fetal bovine serum at 37 °C in a humidified chamber with 5% (*v*/*v*) CO_2_. For experiments on cell proliferation and cytotoxicity after treatment with SOD1 species lacking the incorporated Zn ions, NSC-34 cells (1.5 × 10^3^ cells/well) were harvested in 96-well plates (Falcon, Corning, NY, USA). SOD1 species lacking the incorporated Zn ions, such as apo-WT SOD1^SH^ or apo-SOD1^SH^ that lacks the Zn binding site (H63A/H71A/H80G/D83A), were dissolved in 20 mM 4-(2-hydroxyethyl)-1-piperazineethanesulfonic acid (HEPES, pH 7.2) pretreated with a Chelex 100 Resin (100–200 mesh; Bio-Rad) followed by filtered sterilization, and were diluted in DMEM/F-12 with GlutaMAX medium (Thermo Fisher Scientific) containing 1% (*v*/*v*) non-essential amino acids (Thermo Fisher Scientific) and 1% (*v*/*v*) fetal bovine serum. The cells were exposed to the SOD1 species lacking the incorporated Zn ions at a concentration range from 0.01 to 10 μM for 48 h. In parallel with the exposure of the Zn-deficient SOD1, the cells were also treated with 20 mM HEPES (pH 7.2), which was used as a vehicle control. For experiments on the coordination of Zn ions to the SOD1 species lacking the incorporated Zn ions, 1 μM apo-WT SOD1^SH^ or apo-SOD1^SH^ that lacks the Zn binding site was treated with 1 to 4 molar equivalents of Zn ions at 4 °C for 3 h. The cells were exposed to the SOD1 species treated with Zn ions for 48 h. The cell proliferation and cytotoxicity were evaluated using the Cell Counting Kit-8 and the Cytotoxicity LDH Assay Kit-WST, respectively, in accordance with the manufacturer’s protocol (Dojindo Laboratories, Kumamoto, Japan).

### 4.5. Immunofluorescence

Transgenic mice carrying human mutated G93A SOD1 [B6SJL-Tg(SOD1-G93A)1Gur/J, 002726] or human WT SOD1 [B6SJL-Tg(SOD1)2Gur/J, 002297] were purchased from the Jackson Laboratory (Bar Harbor, MA, USA) [57]. Both transgenic lines were maintained through hemizygotes crossing transgenic males with F_1_ non-transgenic females on a B6SJL background. All animal protocols adhered to the Nihon University Animal Committee guidelines for the care and use of laboratory animals, and were approved by the Institutional Animal Care and Use Committee of Nihon University (#2026).

G93A SOD1 mice were anesthetized with pentobarbital when they reached 60 days (presymptomatic stage), 90 days (symptomatic stage), and 120 days of age (terminal stage), and the mice were transcardially perfused with ice-cold phosphate-buffered saline (PBS, pH 7.4) followed by 4% (*v*/*v*) paraformaldehyde in PBS. The spinal cord was postfixed with 4% (*v*/*v*) paraformaldehyde in PBS (pH 7.4) at 4 °C for 24 h and embedded in paraffin using a Tissue-Tek VIP5 Jr. (Sakura Finetek, Tokyo, Japan). The lumbar spinal cords of the mice (*n* = 3 per genotype per disease stage) were sliced into 6 μm thick sections. These sections were deparaffinized in xylene, rehydrated in ethanol, and rinsed with deionized water. Epitope retrieval was performed using an autoclave for 20 min in the presence of 10 mM citrate buffer (pH 6.0). To inhibit endogenous mouse immunoglobulins, the lumbar spinal cord sections were treated with an M.O.M. blocking reagent (Vector Labs, Newark, CA, USA) at room temperature for 1 h. Non-specific binding of antibodies was blocked with PBS (pH 7.4) containing 10% (*v*/*v*) donkey normal serum (abcam), 3% (*w*/*v*) IgG- and protease-free bovine serum albumin (Jackson Immunoresearch, West Grove, PA, USA), and 0.1% (*v*/*v*) Trion X-100 at room temperature for 1 h. The following primary antibodies were used: rat monoclonal MS785-MS27 cocktail (2.5 μg/mL, FDV-0021A; Funakoshi, Tokyo, Japan), rabbit polyclonal anti-SOD1 EDI (2.5 μg/mL, SPC-206; StressMarq Bioscences, Victoria, BC, Canada), mouse monoclonal anti-NeuN (2.5 μg/mL, MAB377; Merck Millipore), mouse monoclonal anti-glial fibrillary acidic protein (GFAP) cocktail (1 μg/mL, 556330; BD Bioscience, Franklin Lakes, NJ, USA), and rabbit polyclonal anti-ionized calcium-binding adapter molecule 1 (Iba1, 1 μg/mL, 013-27691; Fujifilm Wako Pure Chemical Corporation). As a secondary antibody, Alexa Fluor 405-conjugated donkey anti-rat IgG (5 μg/mL, ab175670; abcam), Alexa Fluor 488-conjugated donkey anti-mouse IgG (5 μg/mL, ab150109; abcam), or Alexa Fluor 594-conjugated donkey anti-rabbit IgG (5 μg/mL, ab150064, abcam) was used. The sections were mounted in ProLong Diamond antifade reagent (Thermo Fisher Scientific). Fluorescence images were acquired using a confocal microscope, Laser Scanning System Z710 (Carl Zeiss, Oberkochen, Germany).

### 4.6. Statistics

The results are given as the mean ± SD. All statistical tests were performed using the Statcel 4 software version 1 (OMS Publishing Inc., Saitama, Japan). After determining the data normality, multiple group comparisons were performed using one-way ANOVA followed by Tukey–Kramer’s post hoc test. Statistical significance was defined as *p* < 0.05.

## 5. Conclusions

The most significant advance of the present study is that it deepens our understanding of how MS785-MS27-reactive misfolded/non-native SOD1 species participate in the pathogenesis of *SOD1*-ALS at the molecular, toxicological, and pathological levels. The antibody cocktail recognizes various misfolded/non-native SOD1 species lacking the incorporated Zn ions. MS785-MS27-reactive SOD1 species cause decreases in the proliferation and survival of cultured cells, and these species are distributed in the spinal motor neurons of the *SOD1*-ALS mouse model. Given that the MS785-MS27 antibody cocktail is commercially available, many researchers could easily begin investigations of misfolded/non-native SOD1 using it.

## Figures and Tables

**Figure 1 ijms-25-05603-f001:**
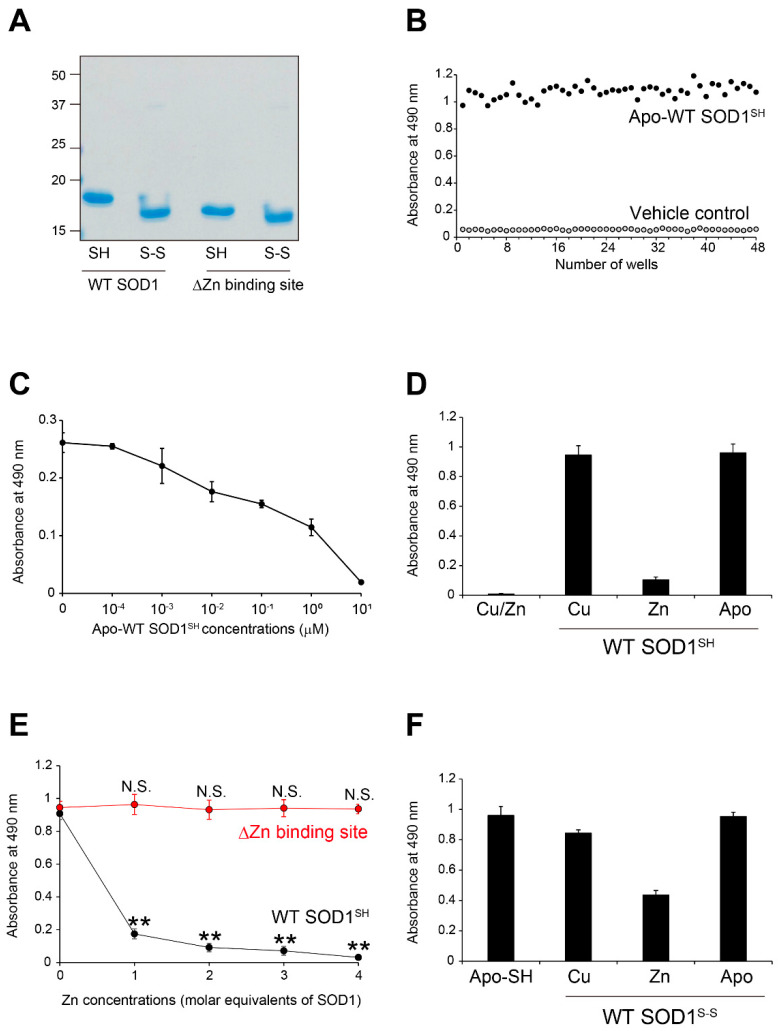
MS785-MS27 antibody cocktail recognizes WT SOD1 species lacking the incorporated Zn ions. (**A**) A representative image showing InstantBlue Coomassie staining with human wild-type SOD1 (WT) and human SOD1 that lacks the Zn binding site by the replacement of amino acids, H63A/H71A/H80G/D83A (ΔZn binding site). Note that SOD1 that lacks the Zn binding site showed faster mobility than WT SOD1. For analysis of SOD1 with the disulfide bond, the protein at 10 μM was treated with 40 mM iodoacetamide at 37 °C for 1 h and was subjected to SDS-PAGE under non-reducing conditions. SH = disulfide bond-cleaved SOD1; S-S = disulfide bond-formed SOD1. (**B**) Specificity and reproducibility of indirect ELISA coupled with the MS785-MS27 antibody cocktail. Apo-WT SOD1^SH^ at 5 μM (black) or vehicle control (Tris-buffered saline with EDTA, gray) was coated onto the 96-well plates. *n* = 48 per group. (**C**) Competitive ELISA for the MS785-MS27 antibody cocktail. The antibody cocktail at 1 μg/mL was preincubated with apo-WT SOD1^SH^ at a concentration range from 10^−4^ to 10^1^ μM. The complex of the antibody cocktail and apo-WT SOD1^SH^ was used as a primary antibody in indirect ELISA. *n* = 6 per concentration. (**D**) Indirect ELISA with the antibody cocktail for WT SOD1^SH^ with different incorporated metal ions. *n* = 6 per group. (**E**) A plot of the ELISA signals of apo-WT SOD1^SH^ or apo-SOD1^SH^ that lacks the Zn binding site treated with 1 to 4 molar equivalents of Zn ions. *n* = 4 per group. The significance of differences was analyzed using one-way ANOVA followed by Tukey–Kramer’s post hoc test. ** *p* < 0.01 (vs. Zn-untreated apo-WT SOD1^SH^). N.S. = not significant. (**F**) Indirect ELISA with the antibody cocktail for WT SOD1^S-S^ with different incorporated metal ions. *n* = 6 per group. All data are given as the mean ± SD.

**Figure 2 ijms-25-05603-f002:**
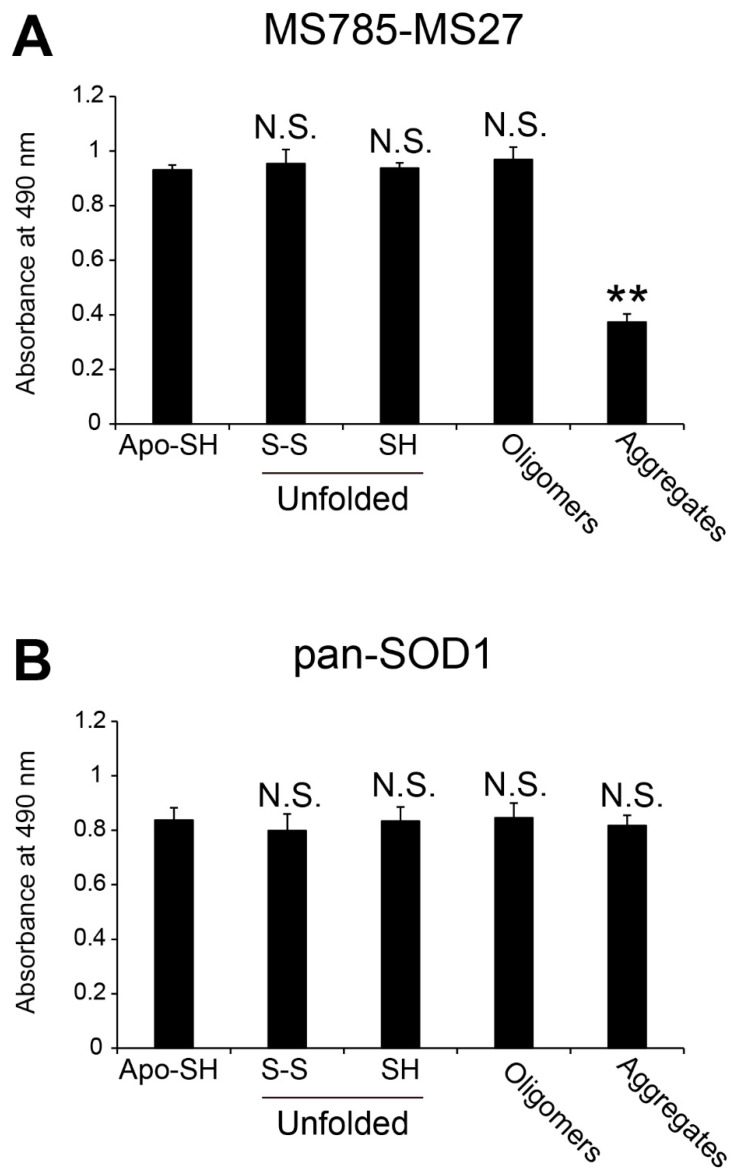
Recognition by MS785-MS27 antibody cocktail of conformation-disordered WT SOD1 species. Conformation-disordered WT SOD1 species, including unfolded, oligomeric, and aggregated forms, were prepared from apo-WT SOD1^S-S^ as a precursor. (**A**,**B**) Indirect ELISA using the conformation-disordered WT SOD1 species analyzed with (**A**) MS785-MS27 antibody cocktail and (**B**) anti-pan SOD1 antibody that reacts with various SOD1 conformers. Apo-WT SOD1^SH^ was used as an internal control for the comparisons among different 96-well plates. All data are given as the mean ± SD (*n* = 6 per group). The significance of differences was analyzed using one-way ANOVA followed by Tukey–Kramer’s post hoc test. ** *p* < 0.01 (vs. apo-WT SOD1^SH^). N.S. = not significant.

**Figure 3 ijms-25-05603-f003:**
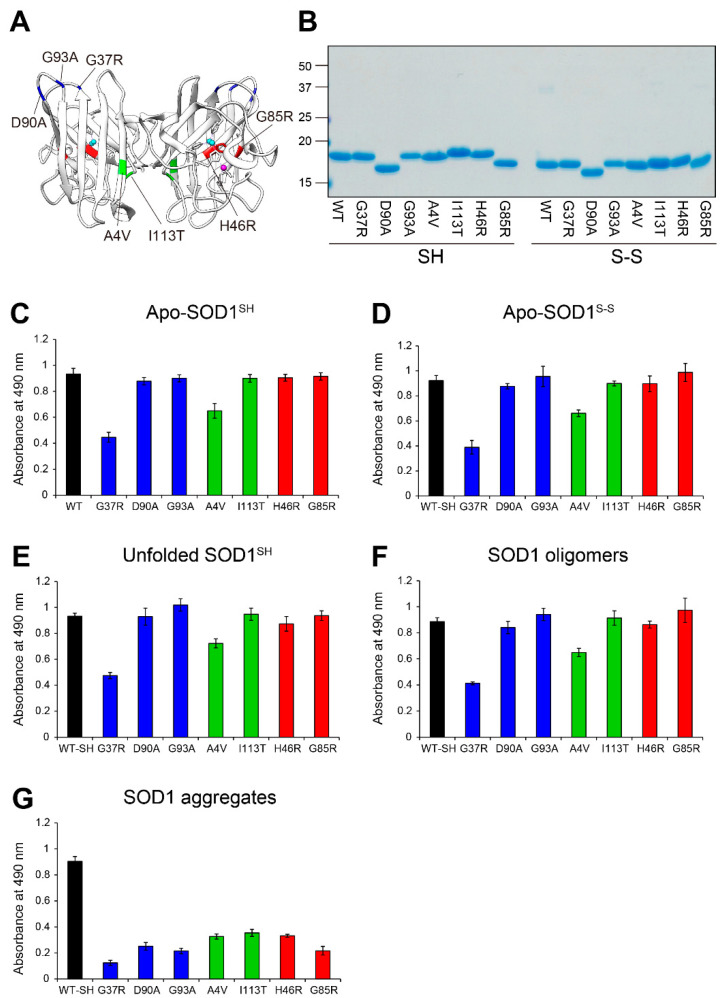
Recognition by MS785-MS27 antibody cocktail of ALS-linked SOD1 mutants with distinct biophysical properties. (**A**) The location of the ALS-linked SOD1 mutations, including A4V, G37R, H46R, G85R, D90A, G93A, and I113T. They are categorized into three groups: (i) wild-type-like mutants (blue; G37R, D90A, and G93A), (ii) dimer interface mutants (green; A4V and I113T), and (iii) metal binding region mutants (red; H46R and G85R). Each mutation is highlighted in the X-ray crystal structure of human WT SOD1 (PDB: 2C9V). Cu and Zn ions are represented by cyan and pink colors, respectively. (**B**) A representative image showing InstantBlue Coomassie staining with WT SOD1 and the ALS-linked SOD1 mutants. For the analysis of SOD1 with a disulfide bond, the protein at 10 μM was treated with 40 mM iodoacetamide at 37 °C for 1 h and subjected to SDS-PAGE under non-reducing conditions. SH = disulfide bond-cleaved SOD1; S-S = disulfide bond-formed SOD1. (**C**–**G**) Indirect ELISA of the recognition by the MS785-MS27 antibody cocktail of (**C**) apo-SOD1^SH^, (**D**) apo-SOD1^S-S^, (**E**) unfolded SOD1^SH^, (**F**) SOD1 oligomers, and (**G**) SOD1 aggregates with the ALS-linked mutations. Apo-WT SOD1^SH^ (black, WT-SH) was used as an internal control. Data are given as the mean ± SD (*n* = 6 per mutant).

**Figure 4 ijms-25-05603-f004:**
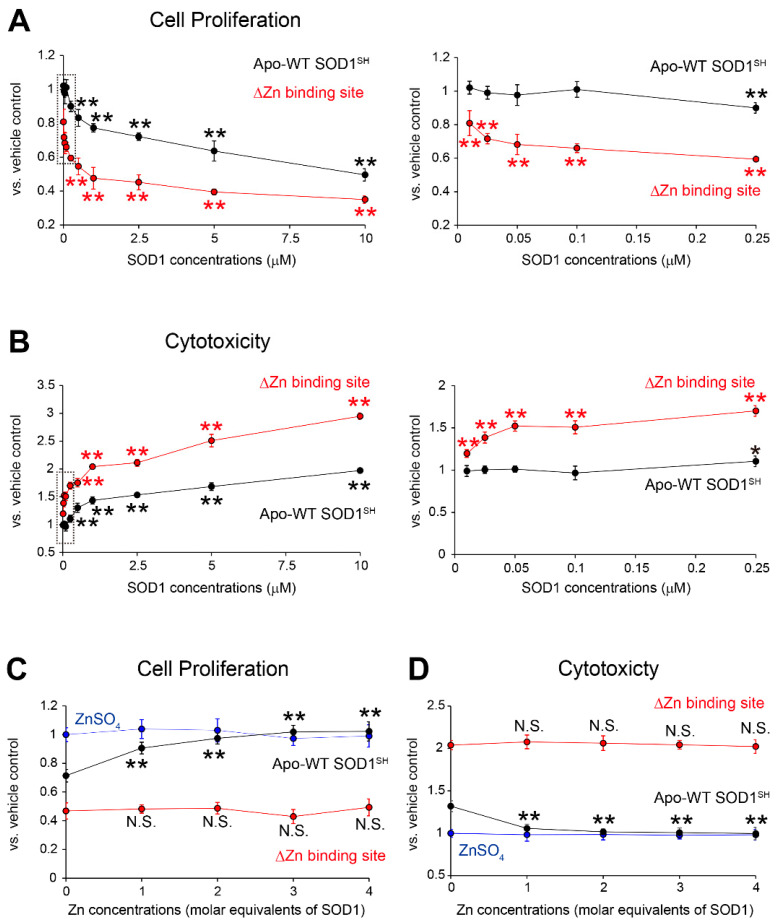
Toxicological features of MS785-MS27-reactive SOD1 species against NSC-34 cells. (**A**,**B**) NSC-34, a cell model of motor neurons, was treated with apo-WT SOD1^SH^ (black) or apo-SOD1^SH^ that lacks the Zn binding site (red) at the indicated concentrations (0.01 μM to 10 μM) for 48 h. HEPES buffer was used as a vehicle control. (**A**) Cell proliferation and (**B**) cytotoxicity were assessed by CCK-8 and LDH assays, respectively. The right panels in (**A**,**B**) represent enlargements of the dotted area in the left panels. The significance of differences was analyzed using one-way ANOVA followed by Tukey–Kramer’s post hoc test. * *p* < 0.05, ** *p* < 0.01 vs. HEPES-treated NSC-34. N.S. = not significant. (**C**,**D**) Apo-WT SOD1^SH^ (black) or apo-SOD1^SH^ that lacks the Zn binding site at 1 μM (red) was treated with 1 to 4 molar equivalents of Zn ions. NSC-34 cells were exposed to the Zn-pretreated SOD1 species. NSC-34 cells were also exposed to ZnSO_4_ resolved in HEPES buffer. (**C**) Cell proliferation and (**D**) cytotoxicity were assessed by CCK-8 and LDH assays, respectively. The significance of differences was analyzed using one-way ANOVA followed by Tukey–Kramer’s post hoc test. ** *p* < 0.01 vs. HEPES-treated NSC-34. N.S. = not significant. All data are given as the mean ± SD (*n* = 6 per group).

**Figure 5 ijms-25-05603-f005:**
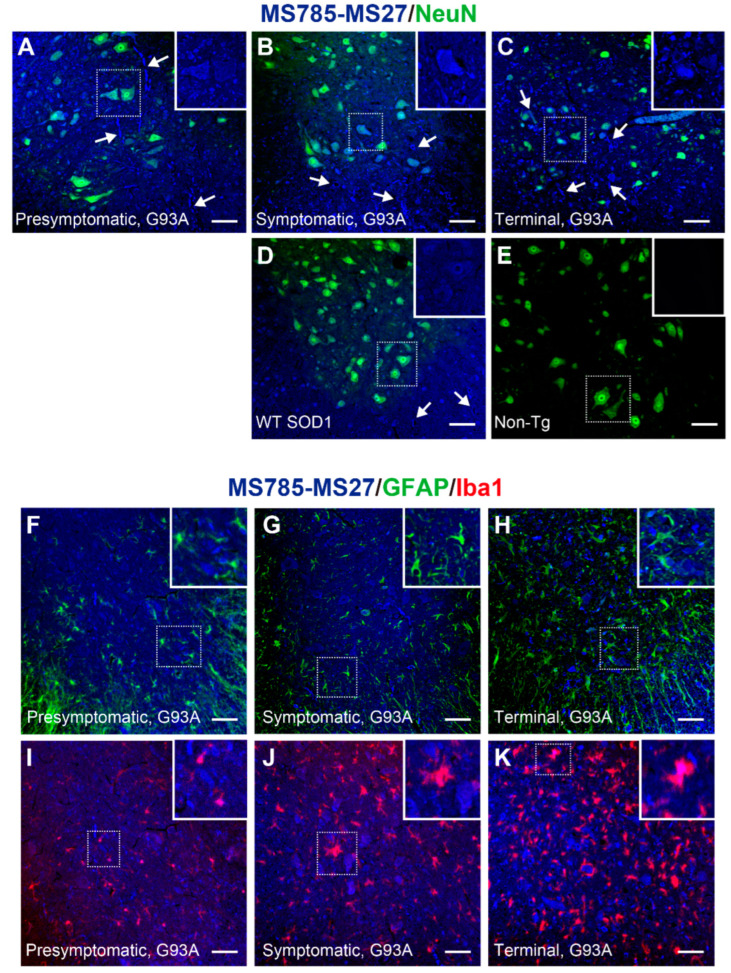
MS785-MS27-reactive SOD1 species are distributed in motor neurons throughout the disease course of G93A SOD1 mice. (**A**–**E**) Confocal imaging of the lumbar spinal cord sections dually immunostained with MS785-MS27 (blue) and NeuN (green), a marker of neurons. G93A SOD1 mice at different disease stages, namely, (**A**) presymptomatic (60 days), (**B**) symptomatic (90 days), and (**C**) terminal (120 days) stages, were used. (**D**) Mice expressing human WT SOD1 and (**E**) non-transgenic (Non-Tg) mice at 120 days of age were also analyzed. *n* = 3 per genotype. Arrows in (**A**) represents axons. Arrows in (**B**,**D**) indicate vacuoles. Arrows in (**C**) stands for fibril-like structures. Insets represent enlargements of the dotted area. (**F**–**H**) Confocal imaging of the lumbar spinal cord from G93A SOD1 mice at (**F**) presymptomatic, (**G**) symptomatic, and (**H**) terminal stages. The sections were immunostained with the MS785-MS27 cocktail (blue) and GFAP (green), a marker of astrocytes. (**I**–**K**) Confocal imaging of the lumbar spinal cord sections immunostained with the MS785-MS27 cocktail (blue) and Iba1 (red), a marker of microglia. G93A SOD1 mice at different disease stages, namely, (**I**) presymptomatic, (**J**) symptomatic, and (**K**) terminal stages, were used. *n* = 3 per disease stage. Scale bars = 50 μm.

**Figure 6 ijms-25-05603-f006:**
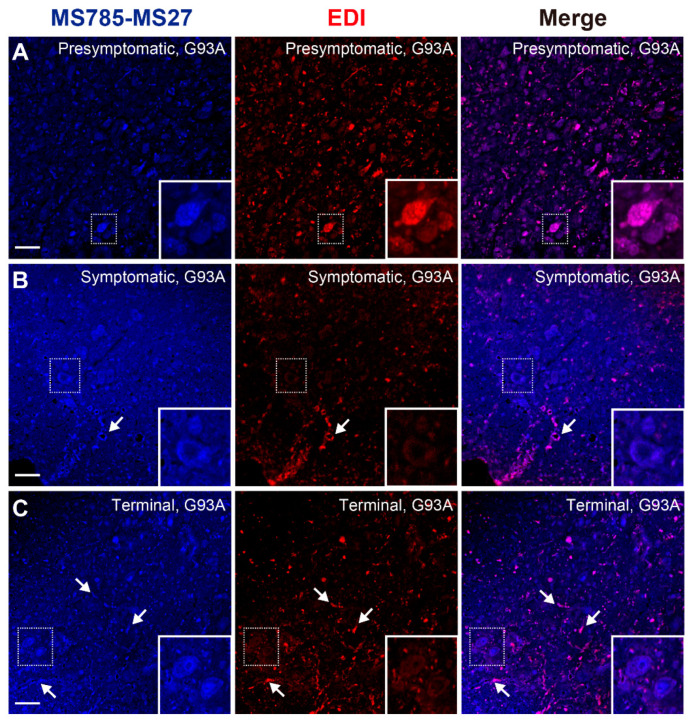
Conformational heterogeneity of misfolded SOD1 species in G93A SOD1 mice revealed by co-immunostaining with MS785-MS27 cocktail and EDI. (**A**–**C**) Confocal imaging of lumbar spinal cord sections from G93A SOD1 mice at (**A**) presymptomatic (60 days), (**B**) symptomatic (90 days), and (**C**) terminal (120 days) stages. The sections were dually immunostained with the MS785-MS27 cocktail (blue) and EDI (red), an antibody specific to misfolded SOD1 species for which the epitope is exposed at the dimer interface. Arrows in (**B**,**C**) represent vacuoles and fibril-like structures, respectively. Insets represent enlargements of the dotted area. *n* = 3 per disease stage. Scale bars = 50 μm.

**Table 1 ijms-25-05603-t001:** ELISA signal intensities of the MS785-MS27 antibody cocktail for apo-SOD1^SH^ with ALS-linked mutations treated with Zn ions.

Apo-SOD1^SH^	Zn Concentrations (Molar Equivalents of SOD1)				
	0	1	2	3	4
WT	1.00 ± 0.051 (100%)	0.22 ± 0.025 ** (22%)	0.11 ± 0.019 ** (11%)	0.08 ± 0.010 ** (8%)	0.05 ± 0.004 ** (5%)
G37R	0.47 ± 0.027 (100%)	0.39 ± 0.018 (83%)	0.34 ± 0.027 ** (73%)	0.30 ± 0.057 ** (63%)	0.26 ± 0.030 ** (55%)
D90A	0.94 ± 0.029 (100%)	0.82 ± 0.031 ** (87%)	0.67 ± 0.006 ** (71%)	0.60 ± 0.005 ** (64%)	0.52 ± 0.025 ** (55%)
G93A	0.95 ± 0.041 (100%)	0.82 ± 0.046 ** (87%)	0.74 ± 0.065 ** (78%)	0.62 ± 0.019 ** (66%)	0.48 ± 0.015 ** (50%)
A4V	0.72 ± 0.015 (100%)	0.66 ± 0.013 ** (92%)	0.63 ± 0.010 ** (87%)	0.59 ± 0.015 ** (82%)	0.54 ± 0.018 ** (75%)
I113T	0.96 ± 0.026 (100%)	0.88 ± 0.008 (92%)	0.82 ± 0.064 * (86%)	0.81 ± 0.029 ** (84%)	0.71 ± 0.053 ** (74%)
H46R	1.02 ± 0.014 (100%)	1.06 ± 0.044 (103%)	1.03 ± 0.041 (101%)	1.02 ± 0.036 (100%)	1.05 ± 0.044 (103%)
G85R	0.98 ± 0.049 (100%)	0.98 ± 0.073 (101%)	1.02 ± 0.062 (104%)	1.02 ± 0.028 (104%)	1.00 ± 0.019 (103%)

All SOD1 variants were used as apo-SOD1^SH^ with or without treatment with Zn ions. The upper dataset represents the absorbance of the ELISA signal intensities, whereas the lower dataset shows the relative change in the ELISA signal intensities compared with the same Zn-untreated SOD1 (set to 100%). Data are given as the mean ± SD (*n* = 3 per mutant). The significance of differences was analyzed using one-way ANOVA followed by Tukey–Kramer’s post hoc test. * *p* < 0.05, ** *p* < 0.01 (vs. the same Zn-untreated SOD1).

## Data Availability

The data presented in this report are available on request from the corresponding author.

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
