# Peer review of "MS785-MS27 Reactive Misfolded/Non-Native Zn-Deficient SOD1 Species Exhibit Cytotoxicity and Adopt Heterozygous Conformations in Motor Neurons"

_ijms, 2024, doi:10.3390/ijms25115603_

Round 1

Reviewer 1 Report

Comments and Suggestions for Authors

The article "MS785-MS27, a Commercially Available Antibody Cocktail against Misfolded SOD1, Recognizes Various Conformation-disordered SOD1 Species Lacking the Incorporated Zn ion" (by Eiichi Tokuda et al.) contains new and useful information about new mechanisms that may be involved in the onset and development of amyotrophic lateral sclerosis. The authors have done extensive work to study the interaction of сommercially available antibody cocktail (MS785-MS27) against misfolded SOD with different forms of this enzyme. The analysis of the work shows that it would be more correct to call the studied forms of SOD "non-native", since the term misfolded has a more limited meaning. For example, it is difficult to agree that after separation of zinc ions or during oligomerization SOD molecules become Misfolded. A shortcoming of the work, in my opinion, is the study of almost only Cocktail MS785-MS27 of the two types of antibodies, except for a few cases. It would have been important to give a comparison of the two antibody types at each stage, so that it would have been clearer how each antibody type contributes to the effects. Some figures are not very informative and could be replaced by a table or simply a statement in the text. For example, Figure 1 B-C and Figure 2. The text should be shortened, avoiding repetition, e.g., in the results and discussion.  It makes sense to add a very brief conclusion.

Reviewer 2 Report

Comments and Suggestions for Authors

Reviewer comments & Suggestions

SOD1 (Cu/Zn superoxide dismutase) is a ubiquitously expressed protein which generally forms homo-dimer. According to many research results, SOD1 is one of the major causative genes of Amyotrophic Lateral Sclerosis (ALS), which is a fatal adult-onset neurodegenerative disease. Around 100 SOD1 mutants were obserevd that are associated with ALS. SOD1 mutant specific monoclonal antibodies were generated for diagnosis of ALS however these mAbs are failed to detect specific SOD mutant ALS. The present research aarticle employed antibody cocktail for detection of SOD mutant associated in vitro as well in vivo experimants.

Recoomndation

1.      Research articles is scientifically sound, interesting for readers and results are presented well and discussed widely.

2.      Data preseneted in figure coorborated with text of result and discussion part.

3.      Refrences are specific and suffcicient.

4.      Methods are described in detail with citation.

Minor comments

5.      What is the major significance of this study? It is well reported that this Antbody MS785/MS27 used in detection of ALS linked SOD mutant form.

6.      Monoclonal antibodies are generated against a specific sequence so how come this 2 antibody cocktails can recognized various mutant form of SOD associated with ALS.

7.      For lines 58-60, provide references.

8.      For Lines 62-66, provide suitable references.

9.      Lines 70-72, provide suitable reference for this.

Reviewer 3 Report

Comments and Suggestions for Authors

The manuscript "MS785-MS27, a Commercially Available Antibody Cocktail 2 against Misfolded SOD1, Recognizes Various Conformation-disordered SOD1 Species Lacking the Incorporated Zn 4 ion" is a search for the cause of the development of ALS.

My comments:

1.       Introduction –

The introduction should be supplemented with genetic variants in the SOD gene, population distributions, and exemplary families with pathogenic variants.

2.       Discussion –

The discussion is missing references to:

How can the authors use the obtained results to identify genetic changes and assess the course of the disease?

Can the obtained results contribute to the introduction of pharmacotherapy in ALS?

3.       A conclusion should be added.

We know about misfolded proteins in Alzheimer's disease. This knowledge did not translate into early diagnosis or improved drug therapy.

What implications might this discovery have for progress in ALS?

Reviewer 4 Report

Comments and Suggestions for Authors

The article is interesting and executed at a high methodological level.

However, the title of the article should be changed. Reading the title of the article immediately gives the impression that the authors are characterizing already characterized antibodies that are commercially available. The title should reflect the essence of the work. The title should focus on pathology, which can be traced from the results of the article.

1) Section 2.1. MS785-MS27 antibody cocktail recognizes WT SOD1 species lacking the incorporated Zn ion in many ways gives the impression of a methodical work. At the end of the section, the authors should make a short summary of the experiments performed.

2) For Figure 2, significant differences between the experimental groups should be reflected

3) Section 2.3. MS785-MS27 recognizes Zn-deficient SOD1 with ALS-linked mutations contains many references to other works, which is not appropriate in such quantity for the research results

4) 2.4. Zn-deficient SOD1 species have cytotoxic effects on NSC-34 cells. Excellent results. Need a more detailed description. The section should be accompanied by a short summary.

5) 2.6. MS785-MS27-reactive SOD1 species are differentially distributed from the known misfolded SOD1 species in G93A SOD1 mice. A fairly voluminous description of the results also requires a brief summary of the results obtained.

6) According to the requirements of the journal, the article requires section 5 – conclusions. Authors should briefly summarize their findings. In this regard, in view of the voluminous materials in the results section, brief summaries under each results section are strictly necessary.

7) In my opinion, the discussion of the results should be expanded and structured by the molecular and cellular effects of the antibody cocktail studied.

Comments on the Quality of English Language

The quality of English is acceptable. However I am not a native English speaker

Round 2

Reviewer 1 Report

Comments and Suggestions for Authors

The authors answered all my questions and made the necessary changes to the article.

Reviewer 4 Report

Comments and Suggestions for Authors

The authors took into account all my comments. The article has been significantly improved and can be accepted in its current form